# Updates in *KMT2A* Gene Rearrangement in Pediatric Acute Lymphoblastic Leukemia

**DOI:** 10.3390/biomedicines11030821

**Published:** 2023-03-08

**Authors:** Mateusz Górecki, Ilona Kozioł, Agnieszka Kopystecka, Julia Budzyńska, Joanna Zawitkowska, Monika Lejman

**Affiliations:** 1Student Scientific Society of Independent Laboratory of Genetic Diagnostics, Medical University of Lublin, 20-093 Lublin, Poland; 2Student Scientific Society of the Department of Pediatric Hematology, Oncology and Transplantology, Medical University of Lublin, 20-093 Lublin, Poland; 3Department of Paediatric Haematology, Oncology and Transplantology, Medical University of Lublin, 20-093 Lublin, Poland; 4Independent Laboratory of Genetic Diagnostics, Medical University of Lublin, 20-093 Lublin, Poland

**Keywords:** *KMT2A*-r, acute lymphoblastic leukemia, infant

## Abstract

The *KMT2A* (formerly *MLL*) encodes the histone lysine-specific N-methyltransferase 2A and is mapped on chromosome 11q23. *KMT2A* is a frequent target for recurrent translocations in acute myeloid leukemia (AML), acute lymphoblastic leukemia (ALL), or mixed lineage (biphenotypic) leukemia (MLL). Over 90 *KMT2A* fusion partners have been identified until now, including the most recurring ones—*AFF1*, *MLLT1*, and *MLLT3*—which encode proteins regulating epigenetic mechanisms. The presence of distinct *KMT2A* rearrangements is an independent dismal prognostic factor, while very few *KMT2A* rearrangements display either a good or intermediate outcome. *KMT2A-*rearranged (*KMT2A*-r) ALL affects more than 70% of new ALL diagnoses in infants (<1 year of age), 5–6% of pediatric cases, and 15% of adult cases. *KMT2A*-rearranged (*KMT2A*-r) ALL is characterized by hyperleukocytosis, a relatively high incidence of central nervous system (CNS) involvement, an aggressive course with early relapse, and early relapses resulting in poor prognosis. The exact pathways of fusions and the effects on the final phenotypic activity of the disease are still subjects of much research. Future trials could consider the inclusion of targeted immunotherapeutic agents and prioritize the identification of prognostic factors, allowing for the less intensive treatment of some infants with *KMT2A* ALL. The aim of this review is to summarize our knowledge and present current insight into the mechanisms of *KMT2A*-r ALL, portray their characteristics, discuss the clinical outcome along with risk stratification, and present novel therapeutic strategies.

## 1. Introduction

Acute lymphoblastic leukemia (ALL) still ranks among the most common childhood malignant hematology diseases, which represents a paramount challenge in 21st-century medicine. This happens because of the fact that continuous chromosomal and molecular abnormalities still evolve and are influenced by genetic and epigenetic processes, as well. However, innovative methods of the in-depth analysis of genetics along with sophisticated experiments tend to have a significant role in assessment, prognosis, and innovative treatment decisions. The attempt to understand, control, and eventually influence some of the unfavorable processes makes genetics nowadays one of the most fascinating, crucial parts of medicine that is also a great inspiration for a broader analysis of the topic of this review. ALL is characterized by the uncontrolled development of large numbers of immature lymphoid cells that lead to the occurrence of the condition [1]. The presence of *KMT2A* rearrangement in ALL is an independent dismal prognostic factor with long-term survival rates of less than 60% across all age groups [2,3,4]. Rearranged *KMT2A (KMT2A*-r*)* ALL presents a complex clinical challenge, with a high incidence in infants and a tendency for aggressive relapse. The *KMT2A* gene (formerly *MLL1*/*MLL*/*ALL*-1/*HRX*/*HTRX1*) is one of the most promiscuous recombination hot spots of the human genome with regard to the onset of malignant diseases. Numerous genomic alterations involving *KMT2A* have been recognized in acute leukemia, including chromosomal translocations, internal tandem duplications, internal deletions, and amplifications. The most common genomic lesions involving *KMT2A* in acute leukemia are chromosomal translocations, resulting in various fusion genes that express an abnormally functioning fusion protein. There are generally 8.3 kb of breakpoint clusters spanning exons 9 to 14 in the *KMT2A* fusion region [5]. Acute leukemia-bearing rearrangements of *KMT2A* have been recognized as a separate entity by the World Health Organization (WHO) since the introduction of the WHO Classification of Neoplastic Diseases of the Hematopoietic and Lymphoid Tissues in 1999 [6]. The current classification recognizes *KMT2A*-r acute lymphoblastic leukemias as B lymphoblastic leukemia/lymphoma with t(v;11q23.3) (*KMT2A*-r) with any fusion partner [7], whereas the updated International Consensus Classification (ICC) of B-acute lymphoblastic leukemia (B-ALL) and T-acute lymphoblastic leukemia (T-ALL) recognizes *KMT2A*-r ALL as B-ALL with t(v;11q23.3)/*KMT2A* rearranged and B-ALL *KMT2A* rearranged-like (in this case as a provisional entity—with frequency <1%, including some with *HOXA* fusions) [8]. The above-mentioned classification (ICC) also considers *KMT2A*-r B-ALL diagnostic considerations and ancillary testing—CD10 nonspecific but characteristically dim/negative, the presence of CD15, the absence of CD24, *KMT2A* break apart in fluorescence in situ hybridization (FISH), and the use of targeted transcriptome sequencing. None of these classifications mention prognostic assessment [8].

The aim of this review is to summarize our knowledge and present current insights into the mechanisms of *KMT2A-*rearranged *(KMT2A*-r*)* ALL, portray their characteristics, discuss the clinical outcomes along with risk stratification, and present novel therapeutic strategies.

## 2. Characteristics of *KMT2A*

The *KMT2A* (also known as mixed lineage leukemia, or *MLL)* is a gene that was described for the first time in 1991–1992 [9,10,11]. *KMT2A* is a large, 90 kb gene containing 36 exons coding for a 431 kDa protein and it is located on the long arm of chromosome 11 band q23.3 (11q23.3). It belongs to the group of *KMT* genes that catalyze the transfer of methyl groups from S-adenosylmethionine to the lysine residues on histone tails, especially H3 [12]. In the case of mutations in one of these genes, an alteration in chromatin conformation occurs along with an incorrect gene expression, leading to several syndromes known as chromatinopathies [13,14,15]. In germline cells, a pathogenic mutation in *KMT2A* leads to haploinsufficiency, resulting in Wiedemann–Steiner syndrome—a rare autosomal-dominant disorder characterized by a delay in development, intellectual disability, unusual facial features, short stature, and hypotonia [16,17,18]. The *KMT2A* gene encodes a DNA-binding protein methylating histone H3 lys4 (H3K4), lysine methyltransferase, formed of 3969 amino acids [9]. This protein has 18 domains, including the SET domain that has the methyltransferase activity on lysine 4 of histone 3 [19]. The main function of the *KMT2A* protein is the epigenetic regulation of transcriptional initiation and elongation through the H3K4 methylation of promoter regions mapped on the target gene [20]. *KMT2A* protein is also responsible for the control of hematopoietic cell proliferation and the differentiation of Meis homeobox 1 (MEIS1) and the homeobox A (HOXA) gene cluster [21,22]. In the case of the deregulation of these genes, the inhibition of correct hematopoietic development triggers the development of leukemia [23]. Rearrangements involving *KMT2A* and its partner genes are found in precursor B-ALL, T-ALL, acute myeloid leukemia (AML), myelodysplastic syndrome (MDS), mixed lineage (biphenotypic) leukemia (MPAL), and secondary leukemia [5]. As far as *KMT2A*-r ALL is concerned, it affects more than 80% of new ALL diagnoses in infants (<1 year of age), 5–6% of pediatric cases, and 15% of adult cases [5,24,25,26]. The above-mentioned classification may also take into account the division into B-cell *KMT2A*-r ALL and T-cell *KMT2A*-r ALL, which is 6% (including 70% of infant ALL cases) and 4–8% of cases, respectively [24,27,28,29,30]. Over 90 *KMT2A* fusion partners have been identified until now, including the most recurring ones—*AFF1* (4q21), *MLLT1* (19p13), and *MLLT3* (9p21)—which encode proteins regulating epigenetic mechanisms and foreshow mostly poor outcomes [21,31]. Fusion between *KMT2A* and *AFF1* is present in about 49% of infants, 44% of the rest of the pediatric population with B-ALL, and almost 75% of adult *KMT2A*-rearranged B-ALLs [31]. The frequency of fusion partners in B-ALL differs depending on patient age, but *AFF1* (formerly AF4) remains the most common fusion partner in all age groups. Several important fusion partners, including their frequency and prognosis in infant and pediatric *KMT2A*-r ALL, are portrayed in Table 1. 

The table presents the correlation between the fusion of *KMT2A* with its partner gene and prognosis (including the frequency). A very poor prognosis relates to a median survival rate of less than 12 months (as in the case of *KMT2A::MLLT1* fusion) [32]. Poor prognosis refers to a median survival rate of 12 to 60 months [33,34,35,36]. *KMT2A*::*MLLT3* fusion manifests an intermediate prognosis in the case of AML with other *KMT2A* translocations [37]. The relationship between *KMT2A* fusion partner and median survival rate in infant and pediatric *KMT2A*-r ALL is shown in Figure 1 and Figure 2.

*KMT2A*-r in ALL usually occurs as a single mutation and does not require a cooperative mutation in order to trigger the leukemia pattern [38,39]. However, it can also be present along with a cooperative mutation, especially often with a PI3K-RAS pathway mutation among infants (14–70%—a wide range with huge differences in reported mutation frequencies [38,39,40,41,42,43,44] and with *KRAS* and *NRAS* mutations among adults (8%) and pediatric patients (26%), respectively) [45]. The significance of intercurrent PI3K-RAS mutation remains the subject of research and further investigation is required. While one study classifies this mutation as a poor prognostic factor, the other ones present no significant impact on the clinical outcome or prognostic features [39,40,42,46], and still others indicate the synergizing effect of PI3K-RAS with *KMT2A* rearrangements that reduces leukemia latency [22,47]. 

Another study showed that *KMT2A* rearrangement often occurs together with the *TP53* mutation (inactivation of the *TP53* tumor suppressor gene). This cooperative mutation concerns ALL as well as MLL but is especially present in infant rearranged leukemias [48]. Another study by A. Stengel concentrated on the investigation of the frequency of the *TP53* mutation among 625 patients with ALL. Thirty-seven patients had *KMT2A*-r ALL while six of them also had a cooperative *TP53* mutation, which translates to 16.2% [49]. Moreover, the study indicates that the incidence of coexisting *TP53* mutation is associated with poor prognosis, especially at relapse.

## 3. *KMT2A*—Clinical Presentation

Infants with *KMT2A-*r present more aggressive features compared with older children [50,51,52,53,54]. The *KMT2A-*r occurs with similar frequency in females and males [55]. *KMT2A*-rearranged (*KMT2A*-r) ALL is characterized by hyperleukocytosis (WBC >30 × 10^9^/L), a relatively high incidence of central nervous system (CNS) involvement, and leukemia cutis (skin infiltration) [50]. Older children, similarly to *KMT2A*-germinal (*KMT2*A-g) newborns, are characterized by lower leukocyte values (WBC < 30 × 10^9^/L), better response to prednisone, and more often positive CD10 [50,55]. They may present with hepatomegaly, splenomegaly, lymphadenopathy, and thrombocytopenia [56]. *KMT2A*-r ALL has a typical immature pro-B immunophenotype. It is characterized by the expression of CD19 and CD34 and the co-expression of CD15, CD33, CD65, and CD68, as well as often a lack of CD10 [46,57,58,59]. It can also have a pre-B immunophenotype, where lymphoblasts are CD22-, CD34-, CD 19-, TdT-, cytoplasmic (Cy) CD79a-, CD10-, and Cy mμ-positive, and cortical/thymic T-ALL, where lymphoblasts are cyCD3-, CD7-, TdT-, CD5-, and CD1a-positive [60]. *KMT2A*-r ALL is also characterized by the specific expression of chondroitin sulfate proteoglycan-4, which is also known as neutron-glial antigen-2 (NG2). It is a transmembrane proteoglycan that is expressed in normal hematopoietic cells very rarely [61,62]. However, the expression of NG2 in the case of *KMT2A*-r ALL is frequent (about 90% of cases) [61]. It contributes to leukemia invasiveness and CNS infiltration and correlates with lower event-free survival (EFS) and more frequent CNS relapse. Because of the predictive value of NG2, it has been the subject of many experiments lately and has also become a new therapeutic target for *KMT2A*-r ALL. The blocking of NG2s quantifies the effect of induction therapy for B-ALL by the transfer of *KMT2A*-r blasts into the blood from the bone marrow, where they are more vulnerable to chemotherapy [63,64,65,66,67]. 

*KMT2A* rearrangement is most common in young children and is associated with a worse prognosis. Laboratory tests of infants with *KMT2A*-r are characterized by a higher leukocyte count and a more severe course with CNS involvement compared to older patients and *KMT2A*-g. From a morphological point of view, there are no ideal criteria for distinguishing the B and T ALL lines. Distinguishing B-lineage lymphoblasts from normal B-lineage lymphoid precursors is also challenging. Pre-B can also express the CD10 antigen, but can be distinguished from mature lymphocytes by their weak expression of CD45 and occasionally the expression of CD34 [60].

## 4. Risk Stratification

There are three major cooperative groups conducting specific clinical trials for infant ALL: Interfant (based in Europe), COG (based in North America), and the Japanese Pediatric Leukemia Study Group (JPLSG). All induction strategies are based on Interfant-99. A prospective risk-stratified approach incorporating *KMT2A*-r *status* and age was used in all recently completed trials (Table 2) [50,54,68,69,70,71,72].

Older patients (>1 year of age) often have a much better prognosis than infants. They have favorable genetic features of high hyperdiploid and *ETV6*::*RUNX1* fusion [59]. Studies have shown that older children share cytogenetic abnormalities with *KMT2A*-g, albeit with a different distribution; the proportion of patients with favorable genetic risk (hyperdiploid, *ETV6::RUNX1*) is higher (60% versus 12% *KMT2A*-g) [55].

Next-generation sequencing (NGS) involves the sequencing of DNA, RNA, or miRNA. It is a tool to identify the most important changes in ALL, which helps to determine the prognosis and pathogenesis of the disease [60,73]. Montaño A. conducted in Salamanca a study on eighty-five patients with B-LL. In cases with *KMT2A*-r, the NGS panel included only the region with the most frequent *KMT2A*-r breakpoints and detected only 70% of cytogenetically confirmed cases, suggesting that patients with a breakpoint at a different location may not be detected by the panel [74]. The performance of NGS in all cases of ALL is thus debatable.

## 5. Clinical Outcome and Interfant Protocol

In chemotherapy, according to the Interfant protocol, low- and high-dose cytarabine (araC) and anthracyclines are the basis. A randomized European study noted that infants with ALL do not have better outcomes with the early intensification of therapy with additional araC and daunorubicin or from the other drugs, mitoxantrone and etoposide [54].

Agraz-Doblas A. et al. analyzed the genome of 124 de novo cases of acute lymphoblastic B-cell leukemia in infants diagnosed and treated according to the Interfant 99/06 protocol. This study used bone marrow or peripheral blood samples from 124 infants <12 months old, who were diagnosed with either pro-B- or pre-B-cell ALL. There were 42 de novo cases, including 27 with *KMT2A*::*AFF1* t(4;11), 5 with the *KMT2A::AF9* (9;11), and 10 without *KMT2A* rearrangement. For validation, an additional cohort of patients comprised 43 with *KMT2A*::*AFF1*, 11 with *KMT2A*::*MLLT3*, and 28 non-*KMT2A* iBCP-ALL cases. Patients with *KMT2A*::*AFF1* were characterized by five-fold longer event-free survival and a three-fold longer overall survival compared to t(4;11) iBCP-ALL patients without *KMT2A::AFF1* [39]. In a randomized study, where patients were treated according to the Interfant-06 protocol, the 6-year event-free survival (EFS) in 651 patients with *KMT2A*-r ALL was 46.1%. For the low-risk group (LR), the EFS was 73.9%, 44.5% for the medium-risk (MR) group, and 20.9% for the high-risk (HR) group. Relapses occurred in 244 (37.5%) patients and the most common (66%) was isolated bone marrow (BM) recurrences [54].

However, in a study of 48 patients treated according to the CCCG-ALL-2015 protocol, *KMT2A*-r B-ALL was reported in 65.51% (19 out of 29) infants and 3.73% (32/857) non-infants, respectively. As a result of treatment, four patients died, and treatment-related mortality (TRM) was 8.33%; 40 patients achieved a complete remission (CR) and the CR rate for the total of 48 patients was 83.33%. Seven patients withdrew from the study. The median follow-up time for the 37 patients without TRM was 15.48 months, 15 patients relapsed, and the 5-year cumulative relapse rate for the 37 patients was 59.16 ± 9.16%. For patients with TRM on (41 patients) or TRM off (37 patients), the 5-year prospective EFS (pEFS) was 36.86 ± 8.48% or 40.84 ± 9.16%, respectively [33].

In a retrospective study of 124 ALL patients over 1 year of age, 31 of whom were with *KMT2A*-r, they were treated from 2008 to 2016 with the GD-ALL-2008 protocol and from 2016 to 2020 with the SCCLG-ALL-2016 protocol. The EFS of *KMT2A*-r-positive children was 56.01 ± 16.89%, and the overall survival (OS) was 73.32 ± 16.6%, which was lower compared to the negative *KMT2A*-r group. In children who received stem cell transplantation in addition to chemotherapy, the 10-year EFS and OS rates in this study were 100% and were higher than in children who received chemotherapy alone; 54.32 ± 16.89% and 72.19 ± 16, 88%, respectively. Unlike other studies, the most common partner gene in their study was *KMT2A*-PTD (Partial Tandem Duplications), which had a 10-year EFS of 85.71 ± 22.37%, showing a good prognosis [75]. 

In a study of 139 patients with ALL, of whom 100 were with *KMT2A*-r and 39 with *KMT2A*-germinal (*KMT2A*-g), 50 children (36%) had CNS involvement. CNS involvement was more common in children with *KMT2A*-r (44 out of 100 patients, 44.0%) compared to children with *KMT2A*-g (6 out of 39 patients, 15.4%). In addition, the frequency of their occurrence depended on the rearrangement of *KMT2A*. Infants with t(9;11)(p21;q23)/*KMT2A::MLLT3* had CNS involvement relatively more frequently (62.5%) than children with t(11;19)(q23;p13.3)/*KMT2A::MLLT1*, in whom the trend was the opposite (only 28.6% of patients). CNS involvement was most common in children <6 months of age. In the group of 50 infants with CNS involvement, as many as 28 had a recurrence (EFS = 0.27, cumulative incidence of relapse (CIR) = 0.65), while among 89 CNS patients—negative—21 experienced a relapse (EFS = 0.58, CIR = 0.26, p < 0.001) for both. In total, 11 of the 49 patients with registered relapses had CNS infiltration (five isolated CNS relapses and six combined CNS and BM relapses) [76]. CNS involvement is more frequently observed in infants than in older children with *KMT2A*-r and is associated with a worse prognosis. Compared to patients with *KMT2A*-g, children with *KMT2A* gene rearrangement are more likely to have CNS involvement.

The Interfant-06 protocol also finds application in treatment with *KMT2A*-germinal (*KMT2A*-g), which is stratified into the low-risk group and usually has a better prognosis compared to *KMT2A*-r. In the 2021 study, which included newborns with *KMT2A*-g ALL treated in accordance with the Interfant-06 protocol, the following results were obtained. The 6-year event-free and overall survival were 73.9% and 87.2%. Relapses occurred early, within 36 months from diagnosis in 28 of 31 (90%) infants. Sites of relapse included (48.4%) isolated bone marrow (BM), (19.4%) isolated CNS, (16.1%) combined BM and CNS, (3.2%) combined BM and testis, and (12.9%) others. Six patients died in CR1, of which five deaths were due to infection. Treatment-related mortality was 3.6%. Age <6 months was a favorable prognostic factor with a 6-year disease-free survival (DFS) of 91% compared with 71.7% in infants >6 months of age (*p* = 0.04). Patients with a high end of induction (EOI) minimal residual disease (MRD) ≥ 5 × 10^−4^ had a worse outcome (6-year DFS 61.4%) compared with patients with undetectable EOI MRD (6-year DFS 87.9%) or intermediate EOI MRD < 5 × 10−4 (6-year DFS 76.4%) [55]. These studies allow us to conclude that in the case of *KMT2A*-g, relapses, if present, tend to occur early in most infants. Treatment-related mortality is very low, and positive prognostic factors include age <6 months and intermediate or negative EOI MRD.

According to our findings, conventional chemotherapy had poor outcomes for pediatric patients with *KMT2A*-r. There is some evidence that allo-HSCT at CR1 might improve the prognosis for patients with risk factors. Infants with *KMT2A*-germline ALL have a good prognosis when they are young at diagnosis and have low EOI MRD.

## 6. Potential Therapeutic Targets

Several potential therapeutic targets that will be discussed in this review are presented in Figure 3.

### 6.1. Histone Deacetylase Inhibitors 

Histone deacetylase inhibitors (HDACi) are enzymes that remove acetyl groups from histones and other proteins, thereby regulating chromatin accessibility and target gene expression (Figure 4) [77,78]. Several studies have shown that HDACi are frequently overexpressed in leukemia cases, leading to the increased expression of tumor-driven genes and abnormal chromatin structure. A limited number of anti-cancer drugs have been approved by the US Food and Drug Administration (FDA). The most researched and prominent HDACi is SAHA [77].

The study by Yao J. and colleagues used an indole-3-butyric acid with phenyl groups in the linker as an HDAC inhibitor. They examined the effect of I1 on HDAC inhibitory activity by determining the level of acetylated histone protein H3 and H4 by Western blotting. Their findings show that the HDAC inhibitor I1, which is a chromatin remodeling factor, has a pronounced anti-proliferative effect on *KMT2A*-r ALL cells by inhibiting cell proliferation by inducing a G0/G1 cell cycle exit. Additionally, I1 inhibits HDAC more effectively than SAHA. I1 inhibited HDAC and also activated the signaling pathway of hematopoietic cell lines when they were treated with I1 at a concentration of 2 μM. Furthermore, I1 inhibited HDAC more effectively in THP-1 cells than in MOLM-13 cells, which is consistent with the IC50 values of I1 in MOLM-13 and THP-1 cells. Accordingly, I1 was able to overcome the cell differentiation block of *KMT2A*-r ALL cells, suggesting potential epigenetic drug potential including in vivo studies and anti-proliferation activity. Furthermore, it would be promising to induce cell differentiation to treat ALL [79].

### 6.2. Curaxin CBL0137

The curaxin family of compounds can activate p53 and inhibit NF-B at the same time. CBL0137 belongs to the second-generation curaxin and shows a very high potential for clinical applications due to the fact that it is soluble in water and has higher metabolic stability in mice compared with other members of the curaxin family [80]. In preclinical in vitro and in vivo models of *KMT2A*-r leukemia, curaxin CBL0137 has antileukemic effects and potentiates the effects of established chemotherapeutic treatments used in the treatment of pediatric high-risk ALL. In studies using CBL0137, nongenotoxic anticancer effects were demonstrated, such as activation of the p53 pathway and the induction of IFN by its chromatin-destabilizing properties [81]. By activating apoptosis and/or causing cell cycle arrest, all antileukemic mechanisms induced by CBL0137 can delay leukemia cell growth, which may have contributed to the *KMT2A*-r ALL growth delay [82].

Based on the evidence of CBL0137’s action in *KMT2A*-r leukemia in vivo, it would be worthwhile to apply this compound with other targeted approaches to *KMT2A*-r ALL that are currently being tested in clinical trials. As a result, the doses of chemotherapeutic drugs can be further reduced and safety may be enhanced. Considering CBL0137’s chromatin-destabilizing effect, one would expect an enhanced therapeutic effect when combined with histone deacetylase inhibitors (HDACi) such as vorinostat or panobinostat. There is evidence that HDACi exerts anti-leukemic effects in vitro, and in particular, LBH589 (panobinostat) shows a promising therapeutic index, since nanomolar concentrations specifically target primary *KMT2A*-r infant ALL cells [83]. Additionally, in a recent study in mouse models, it was observed that panobinostat alone induced the accumulation of DNA damage in the splenocytes of treated mice but did not significantly change the mean leukemia burden or survival. However, the combination of CBL0137 and panobinostat showed the greatest inhibition of leukemia progression and was well tolerated. Curaxin CBL0137 inhibits *KMT2A*-r leukemia cell growth by rapidly inducing apoptosis, and the addition of HDAC increased CBL0137-induced apoptosis. Similar effects were obtained after using CBL0137 in combination with another HDAC inhibitor, entinostat [84,85].

### 6.3. FLT3 Expression 

The fms-like tyrosine kinase 3 (FLT3) is a proto-oncogene expressed on hematopoietic progenitor cells and plays an important role in hematopoiesis. There is a possibility that leukemia can be caused by mutations in the FLT3 receptor [86]. Patients with *KMT2A*-r ALL frequently exhibit constitutive FLT3 activation or increased FLT3 expression resulting from mutations in the tyrosine kinase domain [87]. The use of FLT3 inhibitors has been successful in treating AML with FLT3-activating mutations [88,89].

The Pediatric Oncology Group (COG) AALL0631 investigated whether adding lestaurtinib, a first-generation FLT3 inhibitor, to post-induction chemotherapy improves event-free survival. Following the induction of *KMT2A*-r chemotherapy, infants received chemotherapy alone or chemotherapy with lestaurtinib. The 3-year EFS was not different between chemotherapy plus lestaurtinib (n = 67, 36 + 6%) and chemotherapy alone (n = 54, 39 + 7%, *p* = 0.67) [90]. Despite the lack of the benefit of lestaurtinib, which may be partially due to pharmacological limitations, it illustrates the possibility of testing new targeted therapies in this high-risk group and lays the groundwork for international collaboration [50].

In contrast, FLT3 inhibitors, which potently inhibit FLT3 autophosphorylation, only partially impair the survival of *KMT2A*-r ALL cells. The FLT3 protein can also undergo post-translational modifications, such as glycosylation and ubiquitylation, although it has not been confirmed whether these modifications affect the FLT3 function. The predominant type of protein arginine methyltransferases (PRMTs) is PRMT1, which generates approximately 85% of the asymmetric dimethylarginine (ADMA) proteins. The study revealed that PRMT1 methylates FLT3 at arginine residues at its C-terminus and facilitates the recruitment of signaling adaptor proteins. By modulating FLT3 arginine methylation, PRMT1 contributes to *KMT2A*-r ALL cell survival and growth. Yinghui Zhu et al. consider an important mechanism for the PRMT1-mediated inhibition of FLT3 methylation as a potential treatment for *KMT2A*-r ALL and encourage the further evaluation of MS023 or other potent PRMT1 inhibitors [53].

### 6.4. Menin-KMT2A Inhibitor

Iterative structure-based drug design and X-ray co-crystallography were combined to develop VTP50469, a small molecule inhibitor of Menin–KMT2A interaction [91,92]. Despite the fact that the loss of Menin on the chromatin does not lead to a global loss of *KMT2A* chromatin binding, the *KMT2A*, *DOT1L*, and *SLC43A2* bonds are broken. *DOT1L* chromatin occupancy was probably decreased as a result of the desalination of the DOT1L protein. H3K79me2 levels did not decline globally following VTP50469 treatment because the amount of *DOT1L* destabilized and displaced from chromatin was limited. However, researchers have proven that DOT1L occupancy and H3K79me2 are lost from a subset of *KMT2A* fusion target genes. Moreover, treatment with VTP50469 removed *KMT2A* only from a limited subset of *KMT2A* fusion target genes, which are the same as those losing DOT1L and H3K79me2 occupation [91]. Orally administered as a single agent for 28 days, VTP50469 significantly improved survival and completely eradicated aggressive *KMT2A*-r ALL in a high percentage of mice [91,93].

The inhibition of the Menin–KMT2A interaction in *KMT2A*-r AML and ALL causes similar transcriptional changes as the inhibition of *DOT1L* methyltransferase activity. Comparing Menin-KMT2A inhibition with other methods, the anti-proliferative and differential effects were significantly higher. Therefore, the inhibition of Menin-KMT2A leads to the overall decreased recruitment of ENL and other elongation factors (such as *DOT1L*), which then leads to the observed suppression of *HOXA10*, *MEIS1*, and *MYB* and the upregulation of CD11b [94].

In specific *KMT2A*-fusion target genes (e.g., *MEF2C*, *MEIS1*, *JMJD1C*), concordant gene expression decreases, while a different set (e.g., *HOXA*, *MYB)* is much less vulnerable to Menin-KMT2A perturbation [91,93]. S. Kłossowski et al. developed a highly potent Menin-KMT2A inhibitor MI-3454 which is the second generation of Menin-KMT2A inhibitors with higher potency and optimized drug-like properties. They also found that MEIS1 expression was particularly sensitive to MI-3454 treatment in *KMT2A*-r ALL, identifying MEIS1 expression as a potential pharmacodynamic biomarker for the clinical translation of Menin-KMT2A inhibitors [94]. Given the impressive efficacy observed in the PDX models, these data provide strong support for the development of this approach to clinical evaluation in humans [91].

### 6.5. Proteasome Inhibitors

Proteasomes are intracellular complexes that remove damaged proteins and degrade short-lived regulatory proteins. It is likely that cancer and autoimmune disease may lead to elevated levels of proteasomes in extracellular body fluids. At present, however, little is known about the biological origin and mechanisms of the extracellular transport of these complexes [95].

Glucocorticosteroids (GCs) are the primary component of standard chemotherapy in most acute lymphoblastic leukemia (ALL) regimens; however, *KMT2A*-r ALL is characterized by resistance to GCs. Mousavian Z. et al., in their study, analyzed differential co-expression (DC) networks and protein–protein interactions (PPIs) in *KMT2A*-r infant ALL patients to identify protein modules associated with GC resistance. The results of this work support that proteasome inhibitors and asparagine-depletion drugs can be used as components of the chemotherapy treatment of childhood ALL for patients showing resistance to glucocorticoids [96]. Cheung LC et al. characterized eight infants’ ALL cell lines for immunophenotypic and cytogenetic features. They found that higher doses of a selective proteasome inhibitor, carfilzomib, had a cytotoxic effect against infant cell lines with *KMT2A*-r [97]. In addition to this, Jenkins TW et al., in their study, proved that *KMT2A*::*AF4* cells, the same as T-cell ALL lines, are sensitive to pharmacologically relevant concentrations of specific immunoproteasome inhibitor ONX-0914. Treatment with this inhibitor truly delayed the growth of orthotopic ALL xenograft tumors in mice [98].

### 6.6. Hypomethylating Agents

Hypomethylating agents (HMA) such as cytosine analogs azacitidine (AZA) or decitabine (DEC) inhibit DNA methyltransferases (DNMT) by their incorporation into DNA and by preventing cytosine methylation during cell division. This results in genome-wide demethylation. Both drugs are used in the treatment of acute myeloid leukemia (AML) [99]. 

Roolf C. et al., in a 2018 mouse study of the biological effects of HMA in BCP-ALL with *KMT2A*, analyzed the effectiveness of drugs in mono-application and in combination with conventional cytostatic drugs. It has been shown that HMA reduces the cell proliferation and viability of BCP-ALL. Low-dose drug concentrations were used in combination studies. Researchers have noticed that a combination of low doses of HMA with low doses of cytostatic drugs caused partly stronger anti-proliferative effects compared to the use of a single drug. In addition, it was found that DEC (decitabine) treatment did not eradicate ALL but delayed disease progression in xenograft models [99]. 

Zhang G. et al., in their 2021 study, stimulated human T-cell acute lymphoblastic leukemia molt4 cells with decitabine in vitro and analyzed cell proliferation, apoptosis, and the cell cycle. The results showed that decitabine (especially in low concentrations) reduced viability by inhibiting the PI3K/AKT/mTOR pathway via *PTEN*. When decitabine 50 µM was used instead of 10 µM, *PTEN* expression was downregulated by downregulating *PI3K*, *AKT*, *mTOR*, *P70S6*, and *EIF 4E-binding protein-1*, whereas 10 m upregulated *PTEN* expression by downregulating these enzymes. As a result of decitabine treatment, lipid droplets and autophagosomes were increased. In addition to inducing mitochondrial damage, decitabine inhibits cell proliferation and arrests the G2 phase of the cell cycle. The results of this study emphasize the importance of the appropriate dose of decitabine in treatment [100]. Additionally, a study conducted in 2020 by Schneider P. et al. on mice showed that decitabine moderately delays the progression of leukemia in *KMT2A*-r ALL. It has been reported that low-dose of decitabine with long-term pretreatment lightly sensitizes *KMT2A*-r ALL cells to conventional chemotherapeutics, epigenetic compound-based compounds, and anti-neoplastic agents [101].

### 6.7. Irinotecan

Many cancers have been treated with irinotecan, which is a topoisomerase I inhibitor. Topoisomerase I inhibitors act in a dose-dependent manner, increase enzyme inhibition, and increase cellular topoisomerase levels. Tumor cells are characterized by higher levels of topoisomerase, which makes them more sensitive to irinotecan. Topoisomerase complex with irinotecan prevents the release of topoisomerase and disables the relegation of the nicked strand, which ultimately leads to cell death [102]. 

Kerstjens M. et al. performed in vitro drug screening studies on various models of human ALL cell lines. All ALL cell lines, including those with *KMT2A*-r ALL, were tested with 3685 compounds, and the alkaloid Camptothecin (CPT), at very low concentrations, as well as 10-hydroxycamtothecin (10-HCPT) and 7-ethyl-10-hydroxycamtothecin (SN-38, an active metabolite of irinotecan), had the best clinical effect. After implantation, irinotecan completely blocked the expansion of leukemia in mouse xenografts of a pediatric *KMT2A*-r ALL cell line. Irinotecan monotherapy induced sustained remission in *KMT2A*-r ALL xenotransplanted mice with advanced leukemia [103].

### 6.8. Chimeric Antigen Receptor T-Cell Technology

Chimeric antigen receptor (CAR) T-cell technology is approved for the treatment of relapsed/refractory (r/r) acute lymphoblastic leukemia (B-ALL) and other hematological malignancies [104,105]. The first CAR-T therapy approved for this indication by the Food and Drug Administration (FDA) is tisagenlecleucel (Kymriah) [106]. CAR-T recognizes the B-lymphocyte antigen CD19 and can direct the patient’s T cells to kill CD19+ B-ALL [107,108]. This therapy consists of collecting T cells from the patient, then introducing the CAR construct, and, after lymphodepletion, infusing the modified T cells into the patient [108]. 

Patients with *KMT2A*-r ALL show CD19 antigen loss and immune escape after CAR-T therapy [109]. Therefore, the use of CAR-T therapy in these patients carries the risk of lineage change and relapse in the form of AML, as evidenced by individual cases of patients [105,109]. There is insufficient research on this topic [107].

### 6.9. Hematopoietic Stem Cell Transplantation

Studies show that Hematopoietic stem cell transplantation (HSCT) is more beneficial in infants with *KMT2A*-r ALL with one high-risk (HR) feature: younger age (<6 months old), high white blood cell count (WBC) (≥300,000/μL), or poor response to prednisone [110]. However, due to the late effects of HSCT, some researchers believe that the indication for HSCT should be limited or eliminated in the future [72].

In the Japanese Pediatric Leukemia/Lymphoma Study Group trial MLL-10, by introducing intensive chemotherapy, the indication of HSCT was restricted to patients with high-risk (HR) features only (*KMT2A*-r and either age <180 days or the presence of central nervous system leukemia). There were 56 HR patients, 49 of whom achieved complete remission. In the first remission setting, 43 patients received HSCT. Of those, 38 patients received protocol-specified HSCT with conditioning consisting of individualized doses of busulfan, etoposide, and cyclophosphamide. Overall survival was 80.2% (95% CI, 67.1% to 88.5%) and event-free survival was 56.8% (95% CI, 42.4% to 68.8%) after three years. According to Interfant-06, 14.4% of patients who had HSCT died from HSCT-related toxicity [72].

Researchers Cui Y. and Zhou M. et al. found that allo-HSCT should be recommended for patients with CR1 status. Prospective pEFS was higher in the allo-HSCT group than in the chemotherapy group [51]. The development of new HSCT techniques is necessary in order to avoid acute and late toxicities [111].

## 7. Future Prospects

Our research presents that the combination of curaxin CBL0137 with panobinostat (or another HDAC inhibitor) has a greater inhibition of leukemia progression than the usage of one of the above-mentioned drugs alone. Furthermore, this therapy was well tolerated. The usage of FLT3 inhibitors lays the groundwork for further international collaboration. However, the *PRMT1*-mediated inhibition of FLT3 methylation is a potential treatment for *KMT2A*-r ALL and it is worth further investigation. Proteasome and Menin-KMT2A inhibitors exhibit their efficacy on PDX mice, and still there is a need to continue the research to implement these methods of therapy in humans. The same thing concerns the evaluation of hypomethylating agents (HMA) in mouse studies—a combination of low doses of HMA with low doses of cytostatic drugs caused partly stronger anti-proliferative effects compared to the use of a single drug. Moreover, irinotecan monotherapy induced sustained remission in *KMT2A*-r ALL xenotransplanted mice with advanced leukemia, which gives us a reason to carry out further research on this therapy and, in the event of safety, implement this therapy in humans. (CAR) T-cell technology is an approved therapeutic strategy, but the fact that in some cases it causes lineage switch and relapse as AML needs to be examined more closely. HSCT therapy has a beneficial therapeutic effect, especially on infants with *KMT2A*-r ALL with some high-risk features, but it carries a high risk of negative late effects and HSCT-related toxicity. That is why it is crucial to work on other new HSCT techniques in order to avoid acute and late toxicities.

## 8. Conclusions

In conclusion, *KMT2A*-r ALL in children had moderate remission rates but was prone to relapse with low overall survival and poor outcomes for those treated with chemotherapy alone. For accurate risk stratification, it is important to screen for *KMT2A* partner genes and combine them with other prognostic factors. To conclude, in our review we tried to portray and prove how a thorough understanding of the details can lead us to a holistic view and an understanding of the whole that will finally lead to the achievement of the improvement of the patient’s condition. A careful look at the gene along with the protein it encodes and the processes of fusion with partners allows us to identify pathways that will be targeted for the use of new therapeutic strategies. Better investigation and understanding of genetics move novel therapeutic strategies forward. This review was supposed to be proof of how significantly research in this area can influence the selection of appropriate therapy, response to the treatment, improvement of the clinical condition, and, as a result, improvement of the quality of life and its extension, which is really what medicine in the 21st century is striving for.

## Figures and Tables

**Figure 1 biomedicines-11-00821-f001:**
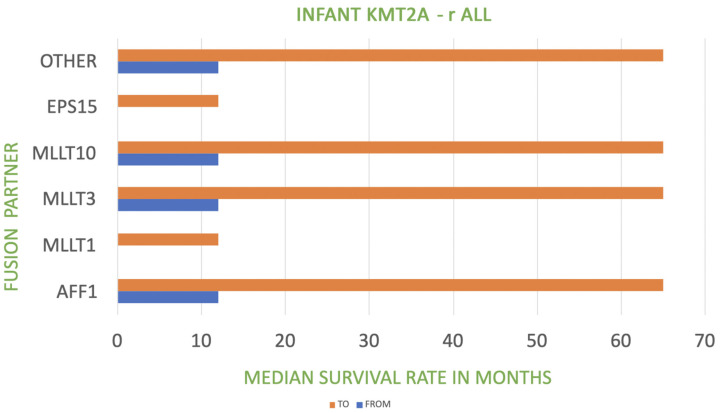
Relationship between *KMT2A* fusion partner and median survival rate in infant *KMT2A*-r ALL.

**Figure 2 biomedicines-11-00821-f002:**
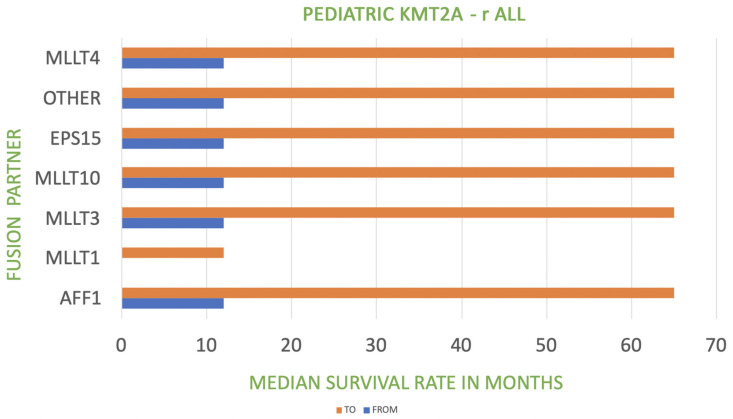
Relationship between *KMT2A* fusion partner and median survival rate in pediatric *KMT2A*-r ALL.

**Figure 3 biomedicines-11-00821-f003:**
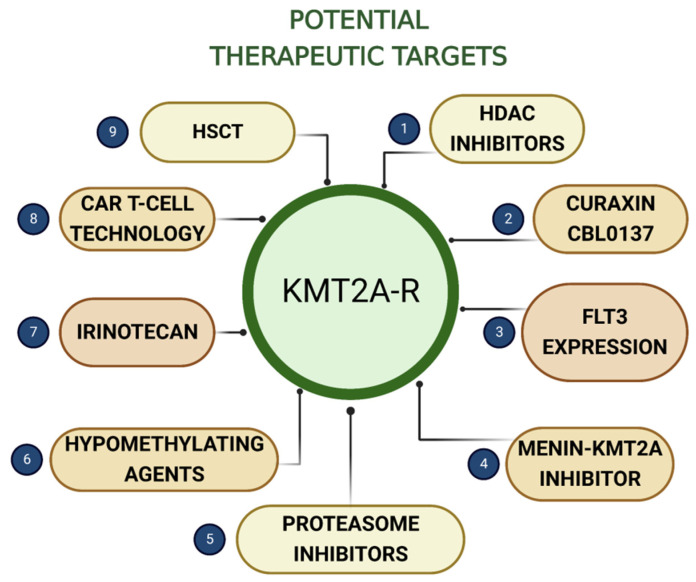
*KMT2A*-r ALL—potential therapeutic targets discussed in the review. HDAC—histone deacetylase; FLT3—fms-like tyrosine kinase 3; CART T-cell—chimeric antigen receptor T-cell; HSCT—hematopoietic stem cell transplantation.

**Figure 4 biomedicines-11-00821-f004:**
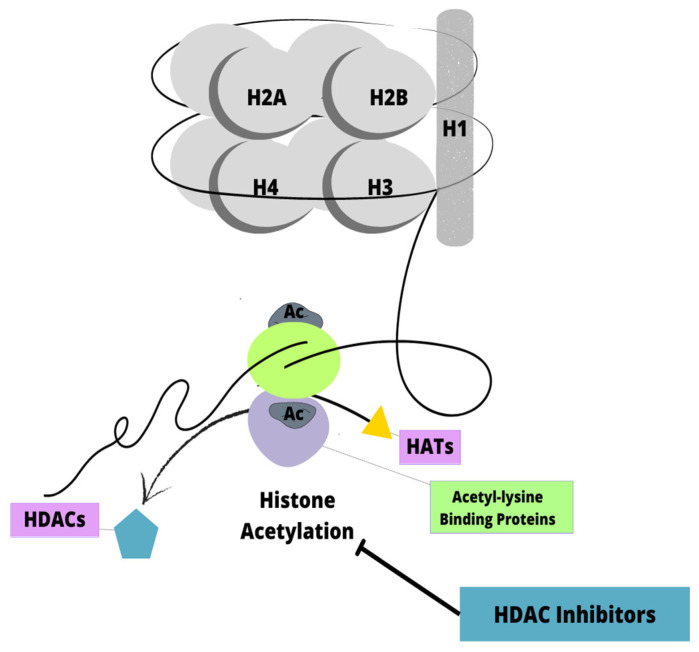
Mechanism of action of HDAC inhibitors. HATs-histone acetylases, HDACs-histone deacetylases.

**Table 1 biomedicines-11-00821-t001:** Examples of common *KMT2A* fusion partners in *KMT2A*-r ALL, with their frequency and prognosis. The frequency rate is based on the study group of 1005 *KMT2A*-r ALL infant and pediatric patients (out of 2345 study groups) [31].

*KMT2A* Fusion Partner in Infant *KMT2A*-r ALL	Frequency	Prognosis
AFF1-4q21	49%	poor
MLLT1-19p13	22%	very poor
MLLT3-9p21	16%	poor to intermediate
MLLT10-10p12	6%	poor
EPS15-1p32	2%	very poor
Other	5%	poor
***KMT2A* Fusion Partner in Pediatric *KMT2A*-r ALL**	**Frequency**	**Prognosis**
AFF1-4q21	44%	poor
MLLT3-9p21	18%	poor to intermediate
MLLT1-19p13	18%	very poor
MLLT10-10p12	5%	poor
MLLT4-6q27	5%	poor
EPS15-1p32	2%	poor
other	8%	poor

**Table 2 biomedicines-11-00821-t002:** Risk stratification in infants with *KMT2A*.

Risk	Interfant	COG	JPLSG	Approximate EFS, %
high	*KMT2A*-r and age <180 days and WBCs ≥ 300,000/μL	*KMT2A*-r and age <90 days	*KMT2A*-r and (age <180 days or CNS leukemia or poor prednisone response)	20
intermediate	other *KMT2A*-r	other *KMT2A*-r	other *KMT2A*-r	50
low	*KMT2A*-g	*KMT2A*-g	*KMT2A*-g	75

## Data Availability

No new data were created or analyzed in this study. Data sharing is not applicable to this article.

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
