# Peer review of "Updates in KMT2A Gene Rearrangement in Pediatric Acute Lymphoblastic Leukemia"

_biomedicines, 2023, doi:10.3390/biomedicines11030821_

Round 1
Reviewer 1 Report
This manuscript, review type, written by Dr. Mateusz Górecki, with the title of "Updates in KMT2A gene rearrangement in pediatric acute lymphoblastic leukemia", made a review of the recent updates of the t(v;11q23.3); KMT2A-rearranged Acute lymphoblastic leukemia (ALL).
Acute lymphoblastic leukemia (ALL)/lymphoblastic lymphoma (LBL) is the most common childhood malignancy. ALL/LBL accounts for approximately one-third of all childhood malignancies and is the most common form of cancer in children; ALL/LBL is five times more common in children than acute myeloid leukemia (AML). The distribution of ALL/LBL categories is B lineage (85 percent), T lineage (10 to 15 percent), and NK lineage (<1 percent).
Infant B-ALL/LBL with KMT2A translocations have a unique gene expression profile, suggesting that it constitutes a distinct disease entity.
The review is well written, with a lot of details.
Comments:
1) Is there any difference in the ALL/LBL classification between the 5th edition WHO classification, and the International Consensus Classification of Myeloid Neoplasms and Acute Leukemias? Any differences for this particular entity in the diagnosis and prognostic assessment?
2) Could you please describe in Table 1 the difference between poor, very poor, intermediate? Any survival curve available?
3) Regarding the mechanism. Is it possible to make a figure showing the intracellular pathways that are being activated/downregulated because of the KMT2A rearrangement, their oncogenic effect, and the target drug (if available)?
4) Could you please add a brief summary of the clinical evaluation, laboratory, and differential diagnosis?
5) Any relationship between KMT2A and mutations of TP53?
6) Should NGS be performed in all cases of ALL/LBL?
Author Response
Response to Reviewer 1 Comments:
Dear Sir or Madam, thank you very much for the review of the manuscript entitled: „Updates in KMT2A gene rearrangement in pediatric acute lymphoblastic leukemia”.
In response to your comments, we would like to thank you for appreciating our manuscript. Thus, you can find paragraphs which involve changes in the corrected manuscript.
We provided the following changes:
- Comment: The discussion about this issue has been added to the Introduction of our review.
- Comment: The differences between different prognosis in Table 1 has been explained. Additionally, there are 2 new charts (Diagram 1&2) portraying the relationship between KMT2A fusion partner and median survival rate.
- Comment: We have drawn up a figure (Diagram 4) regarding the mechanism of action of HDAC inhibitors. It concerns one of the target drugs that show promise in the treatment of ALL with KTM2A rearrangement. TheKTM2A rearrangement mechanism is additionally described in the second section of our review - „Characteristics of KMT2A”.
- Comment: We have summarized the clinical assessment of patients by age and presence of KTM2A-rearranged and KTM2A-germinale, and the possible differential diagnosis of B-lineage and T-lineage ALL and B-lineage lymphoid precursors. The summary can be found at the end of Chapter 3 - „KMT2A - clinical presentation”.
- Comment: The relationship between KMT2A and mutations of TP53 has been described and new references concerning this aspect has been added.
- Comment: The use of NGS in all cases of ALL is debatable. In cases with KMT2A-r, the NGS panel included only the region with the most frequent KMT2A-r breakpoints and detected only 70% of cytogenetically confirmed cases, suggesting that patients with a breakpoint at a different location may not be detected by the panel. We have added a new paragraph about NGS in Chapter 4 - „Risk stratification”.
We honestly hope that our changes will improve the quality of our work. Once again we are very grateful for your review and remain open if you have any other remarks or suggestions that will make our work merit publication in “Biomedicines”.
Reviewer 2 Report
In this review article, the authors summarized our knowledge and presented current insight into the mechanisms of KTM2A-rearranged (KMT2A-r) acute lymphoblastic leukemia (ALL), portray their characteristics, discussed the clinical outcome along with risk stratification, as well as present novel therapeutic strategies.
Comments
This is an interesting review article. The reviewer has some concerns as follows:
1. This manuscript has typos or formatting issues. For examples, (1) there should be space between words: line 50, …HTRX1)is…; line 68, …MLL)is…; line 71, …(11q23.3).It…; line 89, …(MDS),mixed…; and many others; (2) there are missing periods or too many spaces between words: line 60, …1999 [6]…; line 132, …[48-51]. It…; line 144, …to chemotherapy…; and many others; (3) typos: line 183, “15.48 m” changes to “15.48 months”; line 226, “<5 × 10−4” changes to “< 5 × 10−4”; line 347, “[81,83=83,85]” changes to “[81,83,85]”; line 436, “prednisone[100=102]” changes to “prednisone[100,102]”; line 485, “21st" changes to “21st”; (4) what is meaning: line 447, “Researchers Cui Y, Zhou M i in. found…”; line 473, “…, but on the other hand it…”.
2. In Diagram 1, the title and footnotes should be placed below the figure.
3. A paragraph for future perspective can be added at the end of the article.
Author Response
Response to Reviewer 2 Comments:
Dear Sir or Madam, thank you very much for the review of the manuscript entitled: „Updates in KMT2A gene rearrangement in pediatric acute lymphoblastic leukemia”.
In response to your comments, we would like to thank you for appreciating our manuscript. Thus, you can find paragraphs which involve changes in the corrected manuscript.
We provided the following changes:
- Comment: We have checked the entire manuscript and we have made corrections according to the example given.
- Comment: We have made corrections according to the example in Diagram 1. We have applied this rule to all Diagrams and Tables.
- Comment: We have added paragraph number 7 commenting on the future perspective of therapeutic possibilities regarding ALL with KTM2A rearrangement.
We honestly hope that our changes will improve the quality of our work. Once again we are very grateful for your review and remain open if you have any other remarks or suggestions that will make our work merit publication in “Biomedicines”.
Round 2
Reviewer 2 Report
No further comments. This revised manuscript can be accepted.